# The Impact of Producing Type and Dietary Crude Protein on Animal Performances and Microbiota Together with Greenhouse Gases Emissions in Growing Pigs

**DOI:** 10.3390/ani10101742

**Published:** 2020-09-25

**Authors:** Ahmad Reza Seradj, Joaquim Balcells, Laura Sarri, Lorenzo José Fraile, Gabriel de la Fuente Oliver

**Affiliations:** Departament de Ciència Animal, Agrotecnio, Universitat de Lleida, Av. Alcalde Rovira Roure 191, 25198 Lleida, Catalonia, Spain; reza.seradj@udl.cat (A.R.S.); laura.sarri@udl.cat (L.S.); lorenzo.fraile@udl.cat (L.J.F.); gabriel.delafuente@udl.cat (G.d.l.F.O.)

**Keywords:** dietary crude protein, microbiota, greenhouse gases, growing pigs, total tract digestibility

## Abstract

**Simple Summary:**

To study the effect of dietary crude protein (CP) restriction in two different pig producing types and the role of gut microbiota, 32 pure castrated male Duroc and 32 entire male hybrid (F2) piglets were raised in a three-phase feeding regime with a restriction in CP content of the diets. The average body weight of hybrid animals were higher compared to Duroc pigs. No changes were found in average daily feed intake (ADFI) of hybrid animals in comparison to Duroc pigs. Hybrid animals apparently digested more CP than Duroc and Duroc pigs emitted more CH_4_ and ammonia with respect to the hybrids. Dietary protein restriction did not alter emissions of contaminant gases nor microbial community structure in terms of diversity, although some genera were affected by the dietary challenge.

**Abstract:**

In order to reduce dietary nitrogen and achieve an efficient protein deposition as well as decrease N wastage, we challenged the nutrient utilization efficiency of two different producing types in front of a dietary crude protein (CP) restriction and studied the role of the microbiota in such an adaptation process. Therefore, 32 pure castrated male Duroc (DU) and 32 entire male hybrid (F2) piglets were raised in a three-phase feeding regime. At each phase, two iso caloric diets differing in CP content, also known as normal protein (NP) and low protein (LP), were fed to the animals. LP diets had a fixed restriction (2%) in CP content in regards to NP ones throughout the phases of the experiment. At the end of third phase, fecal samples were collected for microbiota analysis purposes and greenhouse gases emissions, together with ammonia, were tested. No changes were found in average daily feed intake (ADFI) of animals of two producing types (Duroc vs. F2) or those consumed different experimental diets (NP vs. LP) throughout the course of study. However, at the end of each experimental phase the average body weight (BW) of hybrid animals were higher compared to Duroc pigs, whereas a reverse trend was observed for average daily gain (ADG), where Duroc pigs showed greater values with respect to hybrid ones. Despite, greater CH_4_ and ammonia emissions in Duroc pigs with respect to F2, no significant differences were found in contaminant gases emissions between diets. Moreover, LP diets did not alter the microbial community structure, in terms of diversity, although some genera were affected by the dietary challenge. Results suggest that the impact of reducing 2% of CP content was limited for reduction in contaminant gases emissions and highlight the hypothesis that moderate change in the dietary protein levels can be overcome by long-term adaptation of the gut microbiota. Overall, the influence of the producing type on performance and digestive microbiota composition was more pronounced than the dietary effect. However, both producing types responded differently to CP restriction. The use of fecal microbiota as biomarker for predicting feed efficiency has a great potential that should be completed with robust predictive models to achieve consistent and valid results.

## 1. Introduction

In the last decade, the global concern about environmental pollution has turned into a hot issue among the pundits. Like in other sectors, the livestock production sector has drawn much attention to greenhouse gas emissions as well as the wastage of unused nutrients through the manure into the soil and water. Nitrogen waste through manure, nitrous oxide and ammonia emissions during storage, and spreading of manure are positively related to nitrogen excretion in both feces and urine [1]. Increased public concern on the livestock environmental footprint led to EU legislation to regulate the potential quota of atmospheric pollution (Integrated Pollution Prevention and Control; IPPC Directive; Directive EU 2016/2284 [2] on the reduction of national emissions of certain atmospheric pollutants), where animal nutrition is considered as a key strategy. Therefore, optimizing nutrient efficiency is essential for the sustainability of swine production systems, especially in a context where the growing demand for food must be met at an affordable cost without compromising environmental integrity [3]. Several authors have already stated the benefits of reducing dietary crude protein (CP) in essential amino acids (EAAs) in balanced diets to growing–finishing pigs in order to decrease nitrogen waste through manure without compromising animal performance and feed efficiency [4,5,6]; however, scarce information is available over its impact on either toxic (NH_4_) or greenhouse (CH_4_ and N_2_O) gas emissions (ranging potentially between 5% and 7%) potential [7,8]. Moreover, due to the high individual variability of animal performances, especially in specific productions other than the lean meat market, there is still great potential to improve efficiency in livestock production systems by better adapting nutrients supply to animals’ requirements, either individually or in groups (breeds or producing types).

The gastrointestinal tract (GIT) of pigs harbors a very complex and dense microbial community that can be altered by diet—ileal digesta reaching the hindgut can modulate both microbial composition and activity, but on the other hand, symbiotic intestinal microbes play a key role in the host adaptation capability to dietary challenges [9]. Nowadays, interest in the existing links between the animal host and its intestinal microbiota has increased exponentially, especially with the objective of optimizing the digestion processes to find predictive biomarkers that lead to the improvement of precision feeding [10] and reduce the environmental impact.

Thus, the main objective in the present study was to explore the potential to reduce production of pollutant gases by improving CP quality, which was done by analyzing the microbial structure and function of the gut microbiota in two producing types of pigs.

## 2. Materials and Methods

All experimental procedures were approved by the Ethics Committee for Animal Experimentation of the University of Lleida (agreement CEEA 02-04/14) and were performed in accordance with authorization 7704 issued by the Catalan Ministry of Agriculture, Livestock, Fisheries, and Food, Spain. The care and use of animals were in accordance with the Spanish Policy for Animal Protection RD 53/2013, which meets the European Union Directive 2010/63 on the protection of animals used for experimental purposes.

### 2.1. Animals, Diets, Experimental Design, and Sampling Procedure

The study was conducted at the Swine Research Center of Catalonia, located in Torrelameu (CEP; Lleida, Spain). Thirty-two pure castrated male Duroc piglets (≈9 week old) with initial body weight (BW) of 25.16 (SD 3.49) kg and 32 entire male hybrids (F2: progeny of [F1: Duroc × Landrace] dams × Pietrain sires) of the same age with initial BW of 22.63 (SD 1.72) kg were purchased from Selección Batallé^®^ (Girona, Spain). Piglets of each genotype (32 each) were divided in 2 groups of 16, each group accommodated in one of four modules based on minimum BW variation, with each receiving differing CP content in their diets, as described below.

Animals of same genotype inside each module were accommodated in four pens (four pigs/pen) based on minimum BW variation to avoid any competition between the animals procuring the feed. Animals followed a three-phase feeding program (phase 1: 9 to 15, phase 2: 16 to 20, and phase 3: 21 to 25 week of age). Diets (described in Table 1) were formulated to be isoenergetic. They contained normal or restricted (a reduction in 2%) level of CP with regards to the recommendations of Fundación Española para el Desarrollo de la Nutrición Animal (FEDNA) [11] for pigs at each feeding phase. Moreover, in order to follow commercial conditions, essential amino acids (EAAs) supply was similar in both diets to avoid any bias that would compromise animal performance in any of both groups. Half amount of each experimental diet was thoroughly mixed with chromic oxide (Cr_2_O_3_) as an indigestibility marker at the rate of 900 mg/kg and given to half of the pens (8 pens in total, 4 per diet).

Individual variations in body weight of the animals along with pen feed consumption were recorded weekly and individual average daily gain (ADG) was calculated as the slope from linear regression of the body weight on feeding days; the individual average daily feed intake (ADFI) was estimated from the weekly pen consumption, considering the number of animals in each pen. In addition, the efficiency of animals to convert feed into body mass was expressed as gain:feed ratio. At the last week of each feeding phase, those animals fed labeled (Cr_2_O_3_) diets underwent fecal spot sampling (≈50 g) by rectal stimulation at intervals of 8 h during 24 h. Representative fecal samples were stored at −20 °C until further digestibility analyses.

For microbiota analysis, an extra fecal sample (≈20 g) at the end of the third feeding phase (week 25 of age) was collected in a separate falcon tube (15 mL). These fecal samples were frozen instantly in liquid nitrogen, transferred to the laboratory, and stored at −80 °C.

Greenhouse gases (GHG; CO_2_, N_2_O, and CH_4_), together with NH_3_ emissions, were analyzed at the end of third experimental phase (week 25 of age) after the digestibility trial. A representative air sampling was obtained from the outside of the fattening installation and the midpoint of the exhaust air outlet in each module using the procedure suggested by Air Movement and Control Association (AMCA) [13] and following Seradj et al. [14]. Once we obtained the air samples inside the modules, simultaneous measurements of CO_2_, NO_2_, NH_3_, and CH_4_ concentration were analyzed using the photoacoustic technique (Innova 1312 Photoacoustic Multigas Monitor, Innova AirTech Instruments, Nærum, Denmark). Emissions were calculated after the correction of the outlet air volume to the standard temperature and pressure following Cao, et al. [15].

### 2.2. Laboratorial Analyses and Calculations

#### 2.2.1. Chemical Composition

Feed samples were analyzed for their chemical composition following the procedures of Association of Official Analytical Chemists (AOAC) [16]. The DM content was determined using an oven at 60 °C for 48 h. The ash content was determined by incineration on muffle furnace at 550 °C for 4 h (ref. 942.05) to determine the organic matter (OM) content and crude protein (CP) was analyzed by the Kjeldahl method (ref. 976.05) and ether extract (EE) using Soxhlet extraction method with diethyl ether (ref. 920.39). The proportion of neutral detergent fiber (aNDFom) was determined according to Van Soest et al. [17] procedures, using α-amylase but not sulphites, and subtracting ashes from the residue, while the acid detergent fiber (ADF) and lignin (ADL) were determined by the method of Goering et al. [18].

#### 2.2.2. Chromium Detection

Fecal spot samples were thawed at 4 °C overnight and gently homogenized inside their collection tube; then, samples (20 g) from the same animal collected at different intervals of a day at each experimental phase were pooled together to make one grab sample per animal per day. Chromium (introduced as chromic oxide to the diets) as an external marker was detected in collected feed and fecal samples after digestion with nitro-perchloric acid (5:1) using de Vega and Poppi [19] methodology, coupled with inductively coupled plasma optical emission spectroscopy (HORIBA Jobin Yvon, Activa family, Kyoto, Japan).

The coefficient of total track apparent digestibility CTTAD of nutrients was calculated using nutrient to marker ratio in the feed and fecal samples as follows:(1)$$\mathrm{CTTAD}=1-\left(\frac{{\left[\mathrm{Cr}\right]}_{\mathrm{intake}}}{{\left[\mathrm{Cr}\right]}_{\mathrm{excreted}}}\times \frac{X_{\mathrm{excreted}}}{X_{\mathrm{intake}}}\right)$$
where X _excreted_ and X _intake_ are the nutrient concentration (g/kg) in feces and in the feed, respectively, [Cr] _excreted_ and [Cr] _intake_ are the concentration (ppm) of chromium in the feces and the feed, respectively.

#### 2.2.3. DNA Extraction and NGS

Fecal samples kept at −80 °C were freeze-dried and the DNA was extracted using a QIAamp DNA Stool Mini Kit (Qiagen Ltd., West Sussex, UK) following the manufacturer’s instructions. The yield and purity of extracted DNA was assessed using a Nanodrop™ (Thermo Scientific, NanoDrop 2000, Waltham, MA, USA), by measuring the absorbance intensity at 260 nm and the absorbance ratio 260/280, respectively. Libraries were prepared using V3-V4 amplicons from the 16s rRNA gene. Sequencing was performed with Illumina Miseq (Illumina, Hayward, CA, USA), generating 902131 paired-end reads. Sequence data were analyzed following the UPARSE pipeline [20]. Taxonomic assignment of the OTUs (operational taxonomic unit) was done using the MG7 program developed by Era7 Bioinformatics [21], which uses cloud computing for the parallel massive basic local alignment search tool (BLAST) similarity analysis to infer both function and taxonomic assignment. MG7 taxonomic assignment was done based on best blast hit (BBH) obtained after searching the nt database (NCBI).

### 2.3. Statistical Analysis

The statistical analysis of the data was commensurate to the design of the study (completely randomized design; CRD) considering pen as the experimental unit. Data were analyzed with SAS (v 9.4; SAS Institute, Cary, NC, USA) using MIXED model procedure, assuming normal distribution of the data.

Producing parameters (BW, FI, ADG, and Gain:Feed) and coefficient of total tract of apparent digestibility of nutrients (DM, CP, and NDF) data were analyzed as follows:Y_ijklmno_ = µ + Ph_i_ + PT_j_ + Di_k_ + (Ph × PT)_l_ + (Ph × Di)_m_ + (PT × Di)_n_ + Ɛ_ijklmn_(2)
where Y is the dependent variable, µ is the mean value, Ph_i_ is the experimental phase (I, II, and III), PT_j_ is the producing type (Duroc and F2), Di_k_ is the diet (LP and NP) along with their possible interactions, and Ɛ_ijklmn_ is the error.

GHG data together with the NH_3_ emissions at the end of the third phase of study were analyzed as follows:Y_ijkl_ = µ + PT_i_ + Di_j_ + (PT × Di)_k_ + Ɛ_ijk_(3)
where Y is the dependent variable, µ is the mean value, PT_i_ is the producing type (Duroc and F2), Di_j_ is the diet (LP and NP) along with their possible interaction, and Ɛ_ijk_ is the error.

Tukey multiple comparison test was applied and significant differences and tendencies were declared at *p* ≤ 0.05 and 0.05 < *p* ≤ 0.10, respectively.

Biodiversity indices (Shannon Wiener [22], Simpson [23], and Richness [24]), Venn diagrams, Spearman correlations between biodiversity and standardized residual values from performance traits, as well as multivariate analysis (CCA, SPLS-DA) were conducted using packages “vegan”, “ade4”, and “mixOmics” from R (v.3.2; R Core Team, Auckland, New Zealand).

## 3. Results

### 3.1. Diet Composition

During the study, a phase feeding strategy was applied dividing the growing–finishing phase into three phases of five weeks (on average) and diets (LP and NP) were formulated based on recommendations published in FEDNA (2013) for animals of different ages. Normal and low CP diets were identical in covering the nutritional requirements of animals, except for CP where low CP diet was intentionally designed with less (≈2%) CP content compared to the normal one.

Table 1 shows the ingredients used in diets and Table 2 reveals the calculated and analyzed composition of each experimental diet. Diets were mainly composed of cereals, where soybean meal was the sole source of protein for the piglets at the initial phase (weeks 9 to 15 of age) and was partially replaced by rapeseed meal during the second and third phase of study. Sugar beet pulp was added (30 g/ kg) to cover the fiber needs mainly in initial diets. Table 2 confirms that the diets of each phase totally covered the need of standardized ileal digestible (SID) of each essential amino acid without any egregious difference in between.

### 3.2. Performance Parameters

A summary of performance parameters is provided in Table 3. As it was conceived, the effect of experimental phase was significant (*p* < 0.01) on all the studied parameters related with performance. No changes were found in ADFI of animals of two producing types (Duroc vs F2) or those that consumed different experimental diets (NP vs. LP) throughout the course of study (*p* > 0.05). In each phase, animals of two producing types were, on average, fed same amount of feed on the daily basis (*p* > 0.05); however, during the last phase, LP diet was consumed more than NP (3.2 vs 2.9 kg/day; *p* = 0.03).

At the entry of animals to the fattening facilities (9 weeks old) the piglets were distributed based on minimum BW variation to yield iso-weighed pens considering producing type (Duroc and F2) and diet (LP or NP). The data provided for the initial BW (Table 3) show the least variation between genotypes (23.3 vs. 24.5 SEM 1.17; Duroc and F2, respectively) and diets (23.9 vs 23.9 SEM 1.14; LP and NP, respectively), which did not differ statistically (*p* = 1.0 for genotype and diet at the initial BW). However, at the end of each experimental phase, the average BW of F2 animals (52.9, 80.5, and 103.8; at the end of P1, P2 and P3, respectively) were higher (*p* < 0.05) compared to Duroc pigs (49.9, 73.6, and 96.5; at the end of P1, P2, and P3, respectively).

Composition of diet (LP or NP) did not influence the BW throughout the feeding phases (*p* > 0.05). Thus, no variations in BW were found between animals fed different diets.

F2 animals showed numerically higher ADG with respect to Duroc ones in all the feeding phases, although differences did only reach statistical significance during the second (weeks 15 to 18 of age) phase (0.99 vs. 0.84 *p* < 0.01; F2 vs. Duroc, respectively). Variations in performance led to a higher overall ADG in animals of F2 with respect to Duroc (0.85 vs. 0.76 for F2 and Duroc pigs, respectively, *p* < 0.01). Protein level in the diet did not influence the ADG of the animals throughout the experiment (*p* > 0.05). Neither producing type of the animals (Duroc and F2) nor CP content of the diet (LP and NP) influenced the efficiency of animals to convert feed into body mass expressed as gain:feed ratio (*p* > 0.05). No interactions were found between producing type of the animal and diet composition (PT × Di) in performance parameters measured during the experiment (*p* > 0.05).

### 3.3. Coefficient of Total Tract Apparent Digestibility (CTTAD) of the Nutrients

Results from CTTAD determination are showed in Table 4. Dry matter digestibility during the third phase of the study (week 22 to 25 of age) was lower, compared with the other two phases (0.934 vs 0.945 and 0.948 *p* < 0.01; P3 vs P2 and P1, respectively). No differences were found in total tract digestibility of DM between Duroc and F2 or between animals fed NP or LP diets (*p* > 0.05). As the CP content recommended for each feeding phase reduced gradually in diets from first to third phase, the coefficient of total tract digestibility of CP decreased accordingly (0.736, 0.701, and 0.683 for P1, P2, and P3, respectively, *p* < 0.01). Besides, in overall, animals of F2 genotype apparently digested more CP than Duroc (0.724 vs 0.689 for F2 vs Duroc, respectively, *p* < 0.05), where no effect of diet was observed for the total tract digestibility of CP. The apparent digestibility of NDF was similar to that of CP and decreased (*p* < 0.01) along the phases of the experiment up to 21.6% at the third phase (0.304 and 0.286 for P1 and P2, respectively). No effects of producing type or CP content of the diet were observed in digestibility of NDF content (*p* > 0.05). Moreover, no interactions between main factors were found in CTTADs (DM, CP, and NDF) measured during the experiment (*p* > 0.05).

### 3.4. Gas Emissions

During the last (week 25 of age; ≈100 kg BW) of the experiment, the emissions (expressed as g per animal per day) of greenhouse gases (CH_4_, CO_2_, and NO_2_) and ammonia (NH_3_), were monitored and the results are tabulated in Table 5. Duroc pigs significantly (*p* < 0.01) emitted (g animal/day) more CH_4_ (10.7 vs 5.6) and ammonia (3.3 vs 0.5) than F2 animals (*p* < 0.05). No differences were observed in NO_2_ and CO_2_ emissions either between diets or between producing types (*p* > 0.05).

### 3.5. Fecal Microbial Characterization

A total number of 7,155,810 reads were obtained, with an average of 119,263 reads per sample. Good’s coverage index resulted in an average of 99.86%, indicating that most of the microbial population present in the samples was covered by the analysis.

All four groups (F2 and Duroc with two level of protein in the diet) shared the 64.13% of the OTUs present in the analysis (see Figure 1) and only an average of 0.54% of the OTUs was specific in each group. Duroc animals showed numerically less specific OTUs than F2 (2.79 vs 3.88%), as well as the animals fed LP diets compared to NP diets (2.59 vs 3.47%). In general, the shared proportion of OTUs was high (96.12, 97.21, 97.41, and 96.53% in Duroc, F2, NP, and LP diets, respectively), indicating a very stable core population in the feces from finishing pigs.

Richness index of diversity showed significant differences between genotypes and diets, with diversity being higher in NP diets and F2 animals (Table 6; *p* < 0.05 in both cases).

Reads belonging to the kingdom Archaea accounted for an average of 0.15% of the total number of sequenced reads and no differences were found in the relative abundance of archaea between either producing types or diets (*p* > 0.05).

Nineteen different phyla were identified within the kingdom Bacteria, although only nine phyla (Actinobacteria, Bacteroidetes, Fibrobacteres, Firmicutes, Fusobacteria, Proteobacteria, Spirochaetes, Tenericutes, and Verrucomicrobia) presented mean abundances above 0.1%. Firmicutes was the most abundant phyla in both genotypes (72.4 ± 6.13% in Duroc and 73.5 ± 4.81% in F2), followed by Bacteroidetes and Proteobacteria. Among main phyla, a genotype effect was found on Bacteroidetes, Proteobacteria, and Verrucomicrobia phyla (Figure 2), with Duroc having higher abundances in the former phylum and lower in rest.

Diet did not affect the microbial composition of the animals at phyla level (*p* > 0.05), but in NP diets, Duroc animals presented a lower proportion of Firmicutes than F2 (interaction effect, *p* < 0.05). Hence, the Bacteroidetes:Firmicutes ratio tended to be lower in Duroc animals (*p* = 0.08, Appendix A).

A description of the main families identified within the four most abundant phyla is shown in Appendix A. Duroc animals showed higher abundances in families *Atopobiaceae* (*p* = 0.007) and *Coriobacteriaceae* (*p* = 0.004) from phylum Actinobacteria, family *Prevotellaceae* (*p* = 0.049) from phylum Bacteroidetes, and family *Lachnospiraceae* (*p* = 0.003) from phylum Firmicutes. On the other hand, F2 animals showed higher abundances in family *Eggerthellaceae* (*p* < 0.001) from phylum Actinobacteria; families *Paludibacteraceae* (*p* = 0.006), *Porphyromonadaceae* (*p* < 0.001), and *Tannerellaceae* (*p* < 0.001) from phylum Bacteroidetes; families *Clostridiaceae* (*p* = 0.003) and *Peptococcaceae* (*p* = 0.001) from phylum Firmicutes; and family *Succinivibrionaceae* (*p* = 0.036) from phylum Proteobacteria (Appendix A).

Diet only affected family *Geobacteraceae* (*p* = 0.001) from phylum Proteobacteria, having higher abundances in animals fed NP diets than those fed LP diets. Interaction effects were present in families *Clostridiales Family XIII, Incertae Sedis* (*p* = 0.038, NP > LP in Duroc animals), and *Eubacteriaceae* (*p* = 0.036, F2 > Duroc in LP diets) from phylum Firmicutes and family *Geobacteraceae* (*p* = 0.019, Duroc animals fed NP diets were higher than the rest of animals) from phylum Proteobacteria (Appendix A).

SPLS-DA analysis enabled the selection of the most predictive or discriminative taxons in the data that helped to classify the samples according to either diet or genotype effect (see Appendix A); from the 10 most predictive taxons, only those with relative abundances higher than 0.01% were considered. In this scenario, five taxons were found to be the most responsible of the differences between genotypes (two varieties of *Holdemanella biformis*, uncultured Coriobacteriales bacterium, uncultured Bacteroidetes bacterium, and uncultured *Prevotellaceae* bacterium) and two related with differences between diets (*Blautia sp.* canine oral taxon 143 and *Selenomonas bovis*).

We compared all the detected phyla and the most abundant bacterial genera (>0.1%, N = 34), with those producing and digestive efficiency parameters studied in the trial (Table 7) using Spearman correlation rank between performance parameters and abundant genera. Only significant correlation values (*p* < 0.05, r > |0.58|) were considered. In Duroc animals, up to nine different genera and two phyla either positively or negatively were correlated with performance (BW, ADG, or digestibility), meanwhile in F2 animals, this correlation was only found in two genera and one phylum. LP diets also presented more correlations between microbiota and performance (five genera and two phyla) than in NP diets (three genera and one phylum).

## 4. Discussion

### 4.1. Effect of Diet

Concern about the environmental pollution caused by the livestock sector has arisen exponentially and this concern involves greenhouse gases (mainly methane and carbon dioxide) emitted and nutrients wasted through the manure on the soil and water. In the present study, we tried to analyze the impact of a dietary challenge as CP reduction in EEAA-balanced diets on both fecal microbiota and environmental impact through pollutant gas emissions. Although we are aware of the use of balance studies to approach this issue, our main aim was to investigate the role of gut microbiota when adapting to CP restriction and their potential impact on the animal host.

Experimental animals were kept under commercial conditions and they were raised in a three-phase feeding regime with the 2% restriction in LP diets that were fixed through the experiment. In agreement with the existing literature [25,26,27,28], no differences were observed in performance between diets (Table 3), and the pigs were able to adjust their metabolism without compromising growth rate. In order to avoid confounding factors, we decided to keep the energy:Lys ratio constant, since an excess in energy in the diet can lead to an increase in the fatness, and thus affect both metabolism and performance [29]. No significant differences were found in contaminant gases emissions (see Table 5) between diets, suggesting that the impact of reducing 2% of CP content was limited. Osada et al. [5] found a reduction in 39% of GHG emissions when reducing a 2.5% CP content in the feed, but animals under their study were growing (38 kg of mean BW, compared with 100 kg of mean BW in our study), and hence more affected by changes in CP content. Lack of response in our study may suggest that our animals already arrived to a steady state in N deposition, although it should be confirmed by N balance studies.

LP diets did not alter the microbial community structure, in terms of diversity, although some genera were affected by the dietary challenge. These results are in agreement with a recent study [30], and suggest the hypothesis that moderate change in the dietary protein levels (2% in our case) can be overcome by at long adaptation of the gut microbiota. However, results do not allow us to get a once for all conclusion in this sense.

More than 95% of total sequences were shared across the animals used in this trial, which suggests a highly predominant and stable core population. SPLS-DA analysis showed *Blautia sp*. canine oral taxon 143 and *Selenomonas bovis* as the most influential taxa in the discrimination by diet. *Blautia* is a genus in the bacterial family Lachnospiraceae that phylogenetic analysis places within the Clostridium coccoides group, also referred to as the Clostridium Cluster XIVa [31]. The common feature of acetogenic *Blautia spp*. may be to become a sink for reductive capability (H^+^) and so alternative pathway to methane synthesis [32]. *Selenomonas spp.* is genus from phyla Firmicutes with a great activity as lactate producer; in both cases, the high proportion of cereals in the LP diet could have influence the proliferation of these types of bacteria.

### 4.2. Effect of Genotype

The authors are aware of the effect of castration on both parameters and the close interaction of breed and sex. However, in the real situation, breeds used to produce commercial male fatty pigs, are submitted to castration, whereas meat hybrids are not. Entire crossbreed animals are focused to produce low-price lean meat, whereas castrated fatty pigs (such as pure Duroc) are used to produce cured products with some specific features such as higher levels of intramuscular fat and precocity. The two extreme producing types allow us to study the animal’s resilience in a nutritional change (i.e., dietary treatments) considering the key role of gut microbiota in such adaption processes. Effort has been done to study microbes’ interspecies interactions rather than titers and/or properties of singular microbes, considering that microbiota interactions may explain relevant aspects of gut functioning in swine, as has been demonstrated recently in human intestine [33,34].

In this study, F2 animals had both higher body weight at slaughter (103.8 vs 96.5, *p* < 0.01) and ADG (0.85 vs 0.76, *p* = 0.01), although gain:feed ratio did not statistically differ between producing types (*p* = 0.57). Compared to other studies with a similar design [30], we could observe a consistency in the observed performance values for F2 animals.

In terms of microbiota profile and among all the diversity indices analyzed, Richness was the only one in which significant differences were found, where F2 animals had a higher number of OTUs in relation to Duroc animals (*p* < 0.05, Table 6). In general, it has been suggested that an increase in microbial diversity is positive in terms of resistance and resilience to dysbiosis and potential pathogens [35]; however, microbial community composition alone does not necessarily provide understanding of community function, since a large degree of functional redundancy exists in the gut microbial ecosystem [36]. Moreover, results depicted in Figure 1 suggest that apart from having differences in diversity, both groups shared a large number of OTUs, indicating a very stable core of microbial population. In a recent study, Holman et al. [37] found a shared proportion among at least 90% of GI samples regardless of experimental variables, including *Clostridium, Blautia, Lactobacillus, Prevotella, Ruminococcus,* and *Roseburia*, which all were found in our study. These genera represent bacteria that are well adapted to the swine gut and may serve as potential markers of a typical swine gut microbiota. No differences were found between producing types when archaea abundance was analyzed (0.12 vs 0.18 % in Duroc and F2 respectively, *p* < 0.05); however, the enteric methane emission was almost 2-fold in Duroc animals compared with F2 (Table 5, 10.7 vs 5.6 g/an/day, *p* = 0.01). Such results are in agreement with the previous ones that indicate the methanogens abundance is not fully correlated with methane production, where certain key species may be responsible of a substantial part of the overall emission [38,39]; however, some influence of the differential size of the hindgut in both productive types cannot be discarded.

Duroc animals presented a lower proportion of *Firmicutes* than F2 in NP diets (interaction effect, *p* < 0.05). Hence, the Firmicutes: Bacteroidetes (F:B) ratio tended to be lower in Duroc animals (*p* = 0.08, Appendix A). Balance between Firmicutes and Bacteroidetes has been recently investigated as an indirect measure of obesity in monogastric animals, including pigs [40]. Distal gut microbiota of obese and lean mice, as well as obese and lean humans, have been compared and the results revealed a statistically significant reduction in the relative abundance of Bacteroidetes and a significant greater proportion of Firmicutes in obese groups than in lean controls [41,42]. Guo et al. [40], also found a significant inverse correlation between body weight and Bacteroidetes abundance. Duroc animals are supposed to have a more active fat metabolism, compared with F2 animals, which are representative of lean pigs. However, to our best knowledge, there is no information regarding F:B ratios in different genotypes. Taking into account that obesity can be considered as a metabolic disease [43], it can be hypothesized that this ratio is only altered in unbalanced situations and not in animals with a more active fat metabolism, that in fact is developed after the nutrients’ absorption [44]. Moreover, it has been described in the literature than in other monogastric animals, such as humans, the age plays an important role in the composition of the gut microbiota. In fact, old age individuals are related with an increase in the abundance of Bacteroidetes [45]. Pure bred Durocs are known to arrive at maturity earlier than commercial crossbreeds [46], and thus, it could be expected a higher contribution of Bacteroidetes in those animals.

### 4.3. Use of Fecal Microbiota as Biomarker for Feed Efficiency

In order to investigate whether the gut microbiota could be related with any of the studied efficiency parameters, we standardized the main performance data (ADG, BW, as well as CP, NDF, and DM digestibility) following a residuals linear regression, including factors such as producing type and initial weight of the animals (see Table 7). In this scenario, we wanted to explore the individual response of the pigs and to see that the differential effect was due to the diet or the producing type. Some recent studies have been trying to get some insight on the links between feed efficiency and gut microbiota in pigs [47,48,49], but the results were inconclusive and more information must be gathered from different breeds and conditions to have an overall overview on the adequacy of using microbiota as biomarker for predicting feed efficiency in pigs.

In the present study, up to 14 genera and five phyla were found to be linked to any of the performance parameters analyzed (Table 7). Fibrobacteres showed positive correlation with efficiency of CP utilization in LP diets and F2 pigs and negative correlation with NDF utilization in LP diets. Metzler-Zebeli et al. [47] found higher abundance of this phyla in highly efficient animals. Although their results were not significant, it can be speculated that a better utilization of CP (especially in diets with CP limitation, e.g., LP diet, and in breeds with higher growth potential, e.g., F2 crossbreed) can be related to an increase in this bacterial phyla. Deferribacteres was positively associated with both ADG (at 110 days) and BW (at 150 days) in LP diets, suggesting a microbial adaptation when CP is limiting. Serino et al. [50], reported changes in Deferribacteres as part of adaptation to a high-fat diet. It can be hypothesized that limiting CP content can unpair the N requirements for the gut microbiota and promote certain groups more involved in N utilization, like Deferribacteres.

Genera including *Prevotella* (negative correlation with CP digestibility in LP diets), *Lactobacillus* (negative correlation with ADG in Duroc animals), and *Ruminococcus* (positive correlation with DM digestibility in Duroc animals) correlated with the performance parameters. Generally, data from Table 7 showed a great influence of some specific bacterial genera over NDF digestibility in Duroc animals (up to four genera, one phyla, and the F:B ratio) and over CP digestibility in LP diets (three genera and one phyla). Other associations should be further investigated.

## 5. Conclusions

To conclude, the influence of the producing type on pollutant gas emissions and fecal microbiota composition was more pronounced than the dietary CP quality effect, which lacked major significant influence. Both producing types appeared to respond differently to CP restriction and quality in terms of microbial composition and gas emissions, but further research should be conducted to confirm these preliminary results. The use of fecal microbiota as biomarker for predicting feed efficiency has a great potential, although it should be completed with robust predictive models.

## Figures and Tables

**Figure 1 animals-10-01742-f001:**
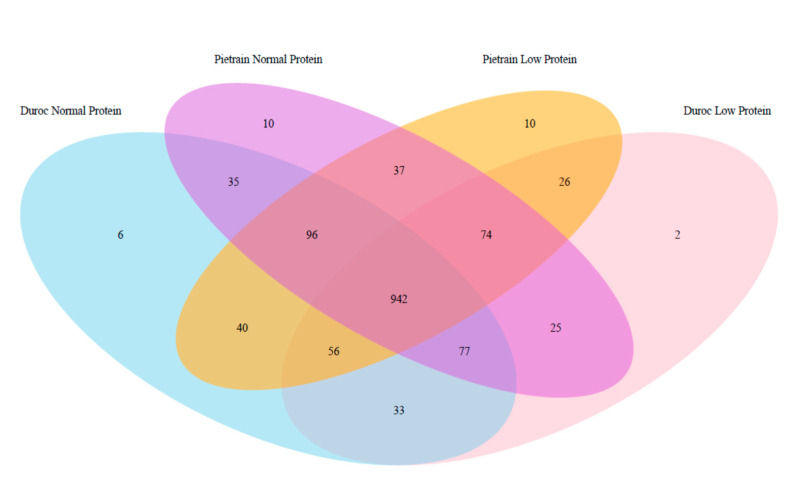
Venn diagram showing the number of shared OTUs (operational taxonomic unit) between animals belonging to Duroc or F2 (Pietrain) producing types and fed with either normal protein (NP) or low protein (LP) diets.

**Figure 2 animals-10-01742-f002:**
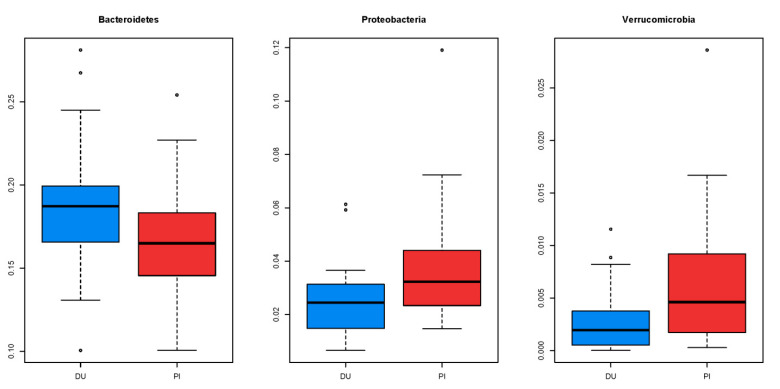
Boxplot of mean abundances of phyla affected by the producing type (DU = Duroc; PI = F2; *p* < 0.05).

**Table 1 animals-10-01742-t001:** Ingredients and additives (g/kg) of the three-phase experimental diets, differing in crude protein content (normal protein, NP vs. low protein, LP) for pigs of 9 to 25 weeks of age.

Ingredients	Feeding Phase ^b^					
	I		II		III	
	LP ^c^	NP ^c^	LP	NP	LP	NP
Barley	181.6	144.4	380.0	120.0	216.7	150.0
Wheat	400.0	400.0	-	300.0	200.0	200.0
Triticale	54.8	32.1	-	100.0	-	121.1
Maize	100.0	100.0	273.9	98.4	250.0	117.1
Bakery byproducts	60.0	60.0	120.0	100.0	110.0	110.0
Rapeseed meal 00	-	-	-	100.0	36.7	100.0
Soybean meal 47% CP	71.0	133.1	118.3	79.6	11.3	23.6
Sunflower meal	-	-	-	-	60.0	56.7
Enersoy 3600	50.0	50.0	-	-	-	-
Rice b	-	-	36.4	50.0	60.0	60.0
Sugar beet pulp	30.0	30.0	8.0	-	-	-
Soybean oil	10.9	14.0	-	-	-	-
Blended animal fat 3/5 acidity	-	-	14.6	19.0	15.1	28.5
Calcium carbonate	12.1	12.0	13.0	10.1	10.7	9.9
Monocalcium phosphate	6.8	6.3	6.5	5.5	3.4	2.3
Sepiolite	-	-	6.1	-	5.0	5.0
Vitamin-mineral premix ^a^	4.0	4.0	5.1	5.1	5.1	5.1
Sodium bicarbonate	-	-	3.2	2.2	2.6	0.5
Sodium chloride	3.8	3.8	4.1	2.1	2.0	2.9
L-lysine, CP 50%	10.0	7.1	6.7	5.9	8.4	6.1
DL-Methionine, 88%	2.3	1.6	1.7	0.7	0.7	0.1
L-Valine	-	-	-	-	0.3	-
L-Threonine	2.4	1.6	1.9	1.4	1.7	0.9
L-Tryptophan	0.3	0.2	0.4	0.1	0.3	0.1

^a^ The vitamin and mineral premix composition is available in Morazán, et al. [12]. ^b^ Phases I, II, and III are 9 to 15, 16 to 21, and 22 to 25 weeks of age, respectively. ^c^ LP: diet with low crude protein content and NP: diet with normal crude protein content based on recommendation of FEDNA [11] for pigs of that age.

**Table 2 animals-10-01742-t002:** Energy and nutrients composition of the experimental diets, differing in CP content (normal, NP vs. low, LP) for pigs of 9 to 25 weeks of age.

SNutrients	Feeding Phase ^d^					
	I		II		III	
	LP ^c^	NP ^c^	LP	NP	LP	NP
Calculated Values ^a^						
ME (MJ/kg)	13.3	13.5	13.0	13.2	13.0	13.2
SID Lysine	10.2	10.1	8.4	8.4	7.6	7.6
SID Lysine/ME (g/MJ)	0.8	0.8	0.6	0.6	0.6	0.6
SID Methionine	4.0	3.7	3.4	2.8	2.5	2.3
SID Methionine + cysteine	6.4	6.3	5.5	5.5	4.6	4.8
SID Threonine	6.5	6.5	5.8	5.8	4.9	4.9
SID Tryptophan	1.8	1.9	1.7	1.7	1.5	1.5
SID Isoleucine	4.9	5.9	4.5	5.0	3.8	4.5
SID Valine	5.7	6.7	5.4	6.0	5.0	5.5
Analyzed Values (g/kg) ^b^						
DM	889.5	892.5	875.8	877.4	879.3	881.5
CP	153.0	173.0	140.0	155.0	126.0	147.0
CF	35.0	36.0	36.0	39.0	46.0	54.0
aNDFom	130.0	120.0	135.0	138.0	141.0	154.0
ADFom	48.0	49.0	45.0	61.0	60.0	70.0
AEE	49.0	57.0	52.0	59.0	46.0	66.0
Starch	432.0	392.0	424.0	406.0	443.0	407.0
OM	929.0	901.0	928.3	930.6	939.3	940.4
p	4.6	4.0	4.6	5.1	4.6	5.1
k	6.0	7.0	7.0	7.0	6.0	7.0

^a^ ME, metabolizable energy; SID, standardized ileal digestible amino acid calculated according to FEDNA [11]. ^b^ DM, dry matter; CP, crude protein; aNDFom, neutral detergent fiber expressed exclusive of residual ash; ADFom, acid detergent fiber expressed exclusive of residual ash; AEE, acid hydrolyzed ether extract; OM, organic matter; p, phosphorous; k, potassium. ^c^ LP, diet with low crude protein content, and NP, diet with normal crude protein content, based on recommendation of FEDNA [11] for pigs of that age. ^d^ Phases I, II, and III are 9 to 15, 16 to 21, and 22 to 25 weeks of age, respectively.

**Table 3 animals-10-01742-t003:** Growth performance (BW, ADG, ADFI, gain:feed) in growing–finishing pigs as affected by dietary CP (normal, NP vs. low, LP) for pigs of 9 to 25 weeks of age.

Parameters ^a^	Genotype		SEM	Diet ^b^		SEM	*p*-Value ^c^			
	Duroc	F2		LP	NP		PT	Di	Ph × PT	Ph × Di
ADFI (Phase I), kg/day	1.3	1.4	0.66	1.4	1.4	0.07	0.92	0.23	0.22	0.01
ADFI (Phase II), kg/day	2.3	2.6	0.66	2.4	2.5	0.07				
ADFI (Phase III), kg/day	3.0	3.0	0.66	3.2 ^e^	2.9 ^f^	0.07				
Overall ADFI, kg/day	2.2	2.4	0.66	2.3	2.2	0.04				
Initial BW (at 9 weeks of age), kg	23.3	24.5	1.17	23.9	23.9	1.14	<0.01	0.70	<0.01	0.08
BW (end of Phase I, 15 weeks of age), kg	49.9 ^b^	52.9 ^a^	0.87	51.6	51.1	0.83				
BW (end of Phase II, 21 weeks of age), kg	73.6 ^b^	80.5 ^a^	1.06	76.0	78.1	1.02				
Final BW (end of Phase III, 25 weeks of age), kg	96.5 ^b^	103.8 ^a^	1.55	100.0	100.4	1.53				
ADG (Phase I), kg/day	0.65	0.70	0.023	0.69	0.67	0.023	0.01	0.74	0.03	0.16
ADG (Phase II), kg/day	0.84 ^b^	0.99 ^a^	0.025	0.90	0.93	0.024				
ADG (Phase III), kg/day	0.78	0.86	0.042	0.84	0.80	0.042				
Overall ADG, kg/d	0.76 ^b^	0.85 ^a^	0.023	0.81	0.80	0.022				
Gain:feed (Phase I), g/g	0.39	0.58	0.120	0.49	0.48	0.018	0.57	0.88	0.06	0.37
Gain:feed (Phase II), g/g	0.22	0.34	0.119	0.28	0.28	0.009				
Gain:feed (Phase III), g/g	0.16	0.26	0.120	0.20	0.22	0.010				
Overall gain:feed, g/g	0.26	0.39	0.119	0.32	0.33	0.008				

^a^ ADFI, average dairy feed intake; BW, body weight; ADG, average dairy gain; Phases I, II, and III are 9 to 15, 16 to 21, and 22 to 25 weeks of age, respectively.^b^ LP, diet with low crude protein content and NP, diet with normal crude protein content based on recommendation of FEDNA [11] for pigs of that age. ^c^ Ph, phase of study; PT, producing type; Di, diet. Different upper case superscripts (a, b) and (e, f) within rows denote differences among producing types and diets, respectively (*p* < 0.05). No interaction (*p* > 0.05) was found between the producing type and the diet (PT × Di).

**Table 4 animals-10-01742-t004:** Coefficient of total tract apparent digestibility (CTTAD) of growing–finishing pigs as affected by CP (normal, NP vs. low, LP) from 9 to 22 weeks of age.

CTTAD ^a^	Phase of Study ^b^			SEM	PT		Diet ^c^		SEM	*p*-Value ^d^		
	I	II	III		Duroc	F2	LP	NP		Ph	PT	Di
DM	0.948 ^a^	0.945 ^a^	0.934 ^b^	0.0020	0.944	0.940	0.944	0.941	0.0018	0.01	0.10	0.10
CP	0.736 ^a^	0.701 ^b^	0.683 ^c^	0.0117	0.689 ^b^	0.724 ^a^	0.703	0.710	0.0095	0.01	0.01	0.60
NDF	0.304 ^a^	0.286 ^b^	0.216 ^c^	0.0164	0.273	0.264	0.281	0.257	0.0134	0.01	0.70	0.20

^a^ CTTAD, coefficient of total tract apparent digestibility of DM, dry matter; CP, crude protein; NDF, neutral detergent fiber.^b^ Phases I, II, and III are 9 to 15, 16 to 21, and 22 to 25 weeks of age, respectively.^c^ LP, diet with low crude protein content and NP, diet with normal crude protein content based on recommendation of FEDNA [11] for pigs of that age. ^d^ Ph, phase of study; PT, producing type; Di, diet. Different upper case superscripts (a, b, c) within rows denote differences among phases of the study and genotype of animals, respectively (*p* < 0.05). No interactions (*p* > 0.05) were found between the producing type and the diet (PT × Di), the phase of study and the producing type (Ph × PT), or the phase of study and the diet (Ph × Di).

**Table 5 animals-10-01742-t005:** The impacts of crude protein content (normal, NP vs. low, LP) on emission of greenhouse gases and ammonia (g/animal/day) at the end of growing–finishing period (ca. 100 kg BW, week 25 of age).

Emission (g/animal/day)	PT		Diet ^a^		SEM	*p*-Value ^b^	
	Duroc	F2	LP	NP		PT	Di
Methane	10.7 ^a^	5.6 ^b^	7.6	8.7	0.53	0.01	0.2
Carbon Dioxide	1496.8	1441.8	1619	1319.6	122.28	0.8	0.2
Nitrous Oxide	0.17	0.09	0.14	0.12	0.026	0.1	0.7
Ammonia	3.3 ^a^	0.5 ^b^	1.7	2.1	0.40	0.01	0.5

^a^ LP, diet with low crude protein content and NP, diet with normal crude protein content based on recommendation of FEDNA [11] for pigs of that age.^b^ PT, producing type; Di, diet. Different upper case superscripts (a, b) within rows denote differences among genotype of animals (*p* < 0.05). No interaction (*p* > 0.05) was found between the producing type and the diet (PT × Di).

**Table 6 animals-10-01742-t006:** Effect of either producing type or diet on the microbial diversity.

Indices	PT		Diet ^a^		SEM	*p*-Value ^b^	
	Duroc	F2	NP	LP		PT	Di
Shannon	3.134	3.178	3.139	3.174	0.0238	0.19	0.36
Simpson	0.894	0.889	0.889	0.894	0.0029	0.18	0.21
Richness	542.4	579.5	575.6	546.2	11.28	0.02	0.07

^a^ LP: diet with low crude protein content and NP: diet with normal crude protein content based on recommendation of FEDNA [11] for pigs of that age. ^b^ PT, producing type; Di, diet.

**Table 7 animals-10-01742-t007:** Spearman rank correlations between standardized residual values and microbiota (*p* < 0.05, r > |0.58|). Positive and negative correlations are highlighted in green and red, respectively.

Phyla		ADG 110d	ADG 150d	ADG 70d	CPd std 130d	DMd std 130d	NDFd std 130d	BW std 110d	BW std 150d
	Genus								
Actinobacteria					Duroc				
Bacteroidetes							Duroc		
Deferribacteres		LP							LP
Fibrobacteres					F2/LP		LP		
Proteobacteria						NP			
*Collinsella*		LP							
*Coriobacterium*					LP			LP	
*Faecalibacterium*			NP				Duroc		
*Fibrobacter*					F2/LP		LP		
*Geobacter*					F2				
*Holdemanella*						Duroc			
*Lactobacillus*				Duroc					
*Mitsuokella*			NP				Duroc		
*Oribacterium*			NP						
*Parabacteroides*							Duroc		
*Prevotella*					LP				
*Roseburia*					Duroc		Duroc		
*Ruminococcus*						Duroc			
*Selenomonas*		LP							
F: B ratio							Duroc		
P: (F+B) ratio						NP			

ADG: average daily gain; CPd: crude protein digestibility; DMd: dry matter digestibility; NDFd: neutral detergent fiber digestibility; BW: body weight.

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
