# Peer review of "The Impact of Producing Type and Dietary Crude Protein on Animal Performances and Microbiota Together with Greenhouse Gases Emissions in Growing Pigs"

_animals, 2020, doi:10.3390/ani10101742_

Round 1
Reviewer 1 Report
Ahmad Reza Seradj et al studied the impact of producing type and dietary crude protein on growth performance and microbiota composition together with greenhouse gases emissions in growing pigs. They suggested that moderate change in the dietary protein levels can be overcome by at long adaptation of the gut microbiota. The topic of their research is interesting. There are several points need to be concerned.
The title: “Microbiota performances” is not appropriate.
Treatment groups were not clearly described in the Abstract.
Present the rational for using only male pigs in the study.
Line 82, 83: units for BW is missing.
Table 3: It is weird that the ADG of phase 3 was lower than that of phase 2. Please explain.
Table 6: Explain “T” and “ns” in the column of P Value.
Author Response
Reviewer 1:
The title: “Microbiota performances” is not appropriate.
[AU]: Title has been changed to animal performances and microbiota
Treatment groups were not clearly described in the Abstract.
[AU]: Description of treatment groups have been now improved in the abstract section.
Present the rational for using only male pigs in the study.
[AU]: The influence of breed and gender on both performance and carcass quality in swine is vastly studied (doi: 10.1017/S1357729800053625). Some breeds have been selected to produce low-cost lean meat, whereas others (i.e. Duroc) are transformed in high-quality products due to their specific features such as higher levels of intramuscular fat (doi: 10.1186/s40104-019-0343-8). The ability of greater incorporation of fat in males, cause their popular use to produce Spanish dry ham production. Moreover, fatty males are conventionally submitted to castration in order to avoid boar taint, whereas leaner breeds which are selected to produce low-cost lean meat usually not.
Line 82, 83: units for BW is missing.
[AU]: Units have been now added to these lines.
Table 3: It is weird that the ADG of phase 3 was lower than that of phase 2. Please explain.
[AU]: The experiment was conducted as a multi-purposes experiment which apart from performance and microbiota trial, the inter-intra muscular fat was studied via small biopsy from the Gluteus medius muscle happened during the third phase of study. Although all animals well recovered from the biopsy trial, authors believe such trivial decrease in ADG of animals during the third phase might be the result of lower intake during few days after practicing biopsy at the phase 3.
Table 6: Explain “T” and “ns” in the column of P Value.
[AU]: Actual P values are now added to the table 6 of the revised manuscript.

Reviewer 2 Report
The stated aim of this paper is to explore the extent to which it is possible to reduce GHG production by reducing dietary protein intake without compromising growth rate and food conversion efficiency. The primary role of dietary protein is to supply essential amino acids (EAAs). In this experiment the LP diet was supplemented with essential amino acids (Tables 1 and 2) so that the intake of EAAs was the same for LP and HP groups at all times. Inexplicably you do not draw attention to this since the formulation of the diets almost ensures that there will be no effect of CP concentration on ADG and feed conversion. What you are demonstrating is that is possible to lower GHG and ammonia excretion by improving protein quality (measured in terms of EAAs). This is not novel.
The differences in ADG between Duroc castrates and F2 entire pigs were predictable from known differences in body composition (protein: fat ratio in carcass). The greater rate of N retention for lean tissue growth in the F2 can explain some differences in N excretion and ammonia production,
The section on the faecal microbiota is novel, potentially important but, as you say of previous studies, inconclusive. In view of this I suggest the two tables and six figures should be reduced to 2-3 tables and one figure (Fig 1). Methane production was significantly greater in the Duroc, suggesting more hindgut fermentation. This could have been associated with a different microbial population in the hind gut but equally, or more likely, due to a larger hind gut having a greater capacity for fermentation. This might have been, but was not, associated with an increase in digestibility, especially NDF. The lower ammonia loss from the F2 pigs could be accounted for by lower N loss in urine associated with a higher lean tissue growth rate. These questions can only be addressed by comprehensive metabolic balance studies.
Several acronyms and terms are undefined or unexplained, (e.g. OTU, SIF). There is no explanation for the descriptions of microbial diversity (Shannon, Simpson, Richness) so I have no idea what they mean.
Author Response
Reviewer 2:
Comments and Suggestions for Authors
The stated aim of this paper is to explore the extent to which it is possible to reduce GHG production by reducing dietary protein intake without compromising growth rate and food conversion efficiency. The primary role of dietary protein is to supply essential amino acids (EAAs). In this experiment the LP diet was supplemented with essential amino acids (Tables 1 and 2) so that the intake of EAAs was the same for LP and HP groups at all times. Inexplicably you do not draw attention to this since the formulation of the diets almost ensures that there will be no effect of CP concentration on ADG and feed conversion. What you are demonstrating is that is possible to lower GHG and ammonia excretion by improving protein quality (measured in terms of EAAs). This is not novel.
[AU]: Indeed, the aim in this study was not to evaluate the effect of the severe restriction of CP in growing pigs under commercial conditions, since animals would have been experienced an impact on their performance, but to lower the CP level without compromising such performance. As it can be seen in acknowledgements section, this study is part of a H2020 European project, based on precision feeding, and above all, we were looking for alternatives that would fit under commercial conditions. However, our hypothesis was that dietary manipulation could have an impact on the hindgut microbial structure and function that lastly might affect both "GHG emission and N losses" as well as the utilization of fermentation products (such as VFA) by the animal host. To clarify all these aspects, we have added paragraphs in L10, L69 and L93.
L10. To study the role of gut microbiota of
L69. Thus, the main objective in the present study was to analyze the impact of CP restriction on the microbial structure and function of the gut microbiota in two producing types of pigs, as well as their eventual impact on performance and contaminant gas emissions.
L93. Moreover, in order to follow commercial conditions, EAAs supply was similar in both diets to avoid any bias that would compromise animal performance in any of both groups.
The differences in ADG between Duroc castrates and F2 entire pigs were predictable from known differences in body composition (protein: fat ratio in carcass). The greater rate of N retention for lean tissue growth in the F2 can explain some differences in N excretion and ammonia production,
[AU]: We agree that by N balance it could be explained some of the variation observed in the animals; however, as it have been showed in previous studies from our group (https://www.feed-a-gene.eu/media/effect-productive-variety-and-dietary-protein-digestive-efficiency-and-fractional-synthesis) breed effect has a major influence that cannot been only explained by protein: fat ratios, but lies in animal’s metabolic efficiency. One of our partial objectives was to investigate whether there was a microbial contribution to this variability, and if it could be measured and predicted in somehow. Other studies are giving more and more importance of the gut microbiota in those processes (doi:10.3390/microorganisms7120622; doi:10.1093/jas/sky060; doi:10.1111/jbg.12433); our aim was to contribute to same subject with research in two producing types overly used in Southern Europe for either lean meat (commercial crossbreed pigs) or more elaborate pig-based products, like cured meat (pure breed Duroc).
The section on the faecal microbiota is novel, potentially important but, as you say of previous studies, inconclusive. In view of this I suggest the two tables and six figures should be reduced to 2-3 tables and one figure (Fig 1).
[AU]: We believe that is important to keep both figures and tables within the text in order to discussed properly the contribution of gut microbiota on the study; however, we do accept to move Figures from 3 to 6 into Supplementary Material to follow reviewer suggestion.
Methane production was significantly greater in the Duroc, suggesting more hindgut fermentation. This could have been associated with a different microbial population in the hind gut but equally, or more likely, due to a larger hind gut having a greater capacity for fermentation. This might have been, but was not, associated with an increase in digestibility, especially NDF.
[AU]: Predicting methane emission from individuals is a difficult task that still now cannot be achieved in its whole. There are multiple factors that influence both qualitatively and quantitatively (reviewed in doi:10.1071/AN17701), and hence we would not speculate that simply a larger hindgut would mean more methane emission. It has been demonstrated that specific species of methanogens contribute significantly to methane emission, and under our knowledge, although it has not been measured the hindgut size on those studies, it does not appear to be “size dependent”, but lies into the early establishment of those microbes into the digestive system. It is true, that a large fermentation chamber would provide more substrate and a slower transit rate that might favor methane emission, but without the main actors in place, methane emission won´t be as considerable as in other individuals. More info in this respect can be found in (doi:10.1017/S1751731110000546; doi:10.1007/BF00394044).
The lower ammonia loss from the F2 pigs could be accounted for by lower N loss in urine associated with a higher lean tissue growth rate. These questions can only be addressed by comprehensive metabolic balance studies.
[AU]: We agree with reviewer that N balance is an appropriate way to asses some of the unknown, but this does not necessary negates other ways to deepen into the efficiency of nutrients utilization and especially the role of gut microbiota on it. This is the major asset we can provide with this study. We believe that the information provided is useful and can be complementary to other studies based on metabolic balances. Nevertheless, we have included a sentence in discussion section (L357) to express the importance of take into account these studies to get a complete picture of the issue.
Several acronyms and terms are undefined or unexplained, (e.g. OTU, SIF). There is no explanation for the descriptions of microbial diversity (Shannon, Simpson, Richness) so I have no idea what they mean.
[AU]: OTU is considered a common acronym for “Operational Taxonomic Unit” in many journals based on molecular biology; however, we have included that explanation in L155 and in Figure 1 legend. We could not find any “SIF” in our manuscript, so no amendments have been made in this sense. We have also included references (L177) to refer the three commonly used biodiversity indexes present in the manuscript. We also recommend for more complete information some further reading, such as doi:10.3389/fmicb.2018.01037 and doi:10.3389/fmicb.2019.02407.

Round 2
Reviewer 2 Report
This draft is little changed from the original. In your accompanying notes you state that this experiment was part of a European study of precision feeding and, in this context, it makes more sense. The diets were designed to provide equal EAA supply at two concentrations of CP; the implication being that the lower CP diet would contain less ‘waste’ protein so contribute less to ammonia and possibly methane production from pig units. This is a sound idea, although hardly novel.
The experiment measured faecal microbiota, which will be representative of the hind gut but not necessarily the terminal ileum. The results are inconclusive. Had there been clear differences attributable to diet, this could have given a lead to future experiments However, the only way to discover the effect of diet (here, the effects of CP surplus to requirement) on ammonia and methane production from pigs, is to measure it. Correlations between differences in hind gut microbiota and differences in production rates could be important.
I believe this study could be acceptable if it was rewritten to stress that the aim was to explore the potential to reduce production of pollutant gases by improving protein quality. The material relating to performance is of no consequence since the effect of phenotype was entirely predictable and the diets were formulated to provide the same amounts of essential nutrients. In this revision you must also concede that your qualitative observations do not permit any definitive conclusions as to differences in hind gut fermentation.
Author Response
[AU]: Attached please find a copy of the second revision of the manuscript. Here we tried to incorporate all the suggestions made by Reviewer 2, especially in two main points. First, we completed the introduction adding more information related to bold the influence of improving protein quality on mitigation of pollutant GHGs (L50-55, L58-65) and consequently we modified the main aim of the study accordingly (L76-78). Second, both discussion and conclusions section were also modified according to reviewer suggestions (see L363-366, L375-382, L386-387, L433-434 and L485-489) highlighting the effect of improving protein quality on contaminant gases and the lack of definitive conclusions on this aspect. Six more references have been added in order to complete and expand the available information.
All changes are highlighted in red throughout the revised manuscript. We hope both Editor and Reviewer will appreciate the effort done in improving the quality of the study following their suggestions and proceed with the final acceptance of the manuscript.
